# Analysis of Long Non-Coding RNA (lncRNA) uc.38 and uc.63 Expression in Breast Carcinoma Patients

**DOI:** 10.3390/genes13040608

**Published:** 2022-03-28

**Authors:** Anna Zadrożna-Nowak, Hanna Romanowicz, Marek Zadrożny, Magdalena Bryś, Ewa Forma, Beata Smolarz

**Affiliations:** 1Department of Chemotherapy, Medical University of Lodz, Copernicus Memorial Hospital, 93-513 Lodz, Poland; anna.m.zadrozna@gmail.com; 2Laboratory of Cancer Genetics, Department of Pathology, Polish Mother’s Memorial Hospital Research Institute, Rzgowska 281/289, 93-338 Lodz, Poland; hanna-romanowicz@wp.pl; 3Department of Surgical Oncology and Breast Diseases, Polish Mother’s Memorial Hospital Research Institute, Rzgowska 281/289, 93-338 Lodz, Poland; zadrozny.lodz@wp.pl; 4Department of Cytobiochemistry, Faculty of Biology and Environmental Protection, University of Lodz, Pomorska 141/143, 90-236 Lodz, Poland; magdalena.brys@biol.uni.lodz.pl (M.B.); ewa.forma@biol.uni.lodz.pl (E.F.)

**Keywords:** lncRNA, transcribed-ultra conserved regions, uc.63, uc.38, breast cancer

## Abstract

Background. The role of the transcribed ultra-conserved regions (T-UCRs) has not yet been fully discovered, but the studies showed some indications that impaired expression of T-UCRS were present in malignant tumors, including breast cancer. Aim. The presented work assessed the expression of two transcribed-ultra conserved regions–uc.63 and uc.38–in breast cancer tissue samples. Material and methods. The research was carried out on a group of 100 patients with invasive ductal carcinoma and 100 patients (test group) with benign tumors in breast tissue (control group). Results. As a result of the statistical analysis, it was shown that the expression of uc.63 and uc.38 is statistically significant, and, accordingly, higher (*p* < 0.0001) and lower (*p* < 0.0001) in the test group than in the control group. Statistical dependency analysis of the expression of uc.63 and uc.38 and the selected clinical and pathological factors showed that the expression of uc.63 statistically drops with the patient’s age (*p* = 0.04), and is higher in the breast cancer tissue type M1 according to the TNM classification (*p* = 0.036) and in tissues with overexpressed HER2 (*p* = 0.035). Conclusion. The obtained results of the statistical analysis indicate a relationship between the expression of uc.63 and uc.38 and the occurrence of breast cancer.

## 1. Introduction

Breast cancer remains the most commonly diagnosed malignant tumor in women in the world. It is also the leading cause of death from malignant tumors among women [1,2,3,4,5,6,7]. Scientists are still looking for modern therapeutic methods, as well as predictive and prognostic factors effective in the fight against the discussed malignant tumor [8,9,10]. Recently, molecules from the rich RNA family have been of particular interest [11,12].

After a long period of research on the function of microRNAs, long non-coding RNA (lncRNAs) have now become the subject of numerous analyses [13,14]. Studies have shown links between impaired lncRNA expression and the pathogenesis of many diseases, including cancer [15,16,17]. lncRNAs transcribed from ultra conservative regions (T-UCRs) of the genome should be distinguished [18].

A number of publications indicate a correlation between the altered expression of these transcripts and the occurrence of malignant tumors, including breast cancer [19,20,21]. However, it has not yet been determined how T-UCRs contribute to the initiation of carcinogenesis. PubMed presents only a few articles evaluating the link between ultra-conservative regions and their transcripts and breast cancer. Four of these studies investigated the relationship between the presence of single nucleotide polymorphisms (SNPs) in ultra-conservative regions and other lncRNA sequences and the risk of breast cancer and the risk of familial occurrence of the malignant tumor in question. The first of these works was published in 2008 by Yang et al. [22]. In their study, the authors found that of the six SNPs evaluated in UCRs, the significance of one polymorphism rs9573903 was borderline significant, and the presence of rs2056116 polymorphism was associated with a significantly higher risk of familial breast cancer, especially in patients under 50 years of age [23]. The study was conducted in the German population in breast cancer patients without *BRCA1/BRCA2* mutations. A year later, an analysis by Catucci et al. was published, in which the importance of the above two polymorphisms in the Italian population was evaluated, also in patients without *BRCA1/BRCA2* mutations [23]. However, this study did not show a relationship between these SNPs and the familial incidence of breast cancer [23]. A negative study was also a study by the Chinese, which analyzed the relationship of seven SNPs present in ultra-conservative regions with the risk of developing breast cancer [24].

In the 2020 work by Suvanto et al., not only was the effect of SNPs in lncRNA genes on breast cancer risk investigated, but it also looked at whether there was a correlation between the occurrence of SNPs in a specific lncRNA gene and the level of its expression [25]. The study authors used data from the Breast Cancer Association Consortium ‘BCAC’ genome-wide association study (GWAS), OncoArray, and iCOGs to select SNPs located in or in close proximity to ultra-conservative regions and lncRNA regions that may be associated with an increased risk of breast cancer. The authors determined the significance of selected SNPs using the INQUISIT and eQTL statistical tools. As a result of the described analysis, an increased risk of breast cancer was shown for SNPs present in one lncRNA, GABPB1 and AS1, and two ultra-conservative regions—uc.184 contained in the CPEB4 gene and uc. 313 in the TIAL1 gene [25]. The authors of the study suggested that the presence of SNPs may have been associated with a change in the expression of transcripts of the above regions, which, among other things, through correlation with *BRCA1*, could affect the development of breast cancer [25]. However, this hypothesis requires further evidence. The effect of altered lncRNA expression on the prognosis in breast cancer has been proven in numerous studies that have been meta-analyzed by Tian et al. [26]. It evaluated 48 non-coding transcripts present in the tissues of the malignant tumor in question or in the blood of patients. Overexpression of three of them, CCAT2, MALAT1, and NEAT1, was associated with shorter overall survival (OS), overexpression of two, CCAT2 and HOTAIR, with a shorter metastatic-free time (MFS), overexpression of another 17 lncRNAs (BCAR4, HOTTIP, CCAT1, Z38, TUNAR, CRNDE, HULC, MVIH, TP73-AS1, linc-ITGB1, PVT1, UCA1, OR3A4, DANCR, LINP1, SNHG15, SUMO1P3) with a worse prognosis overall [26]. In turn, overexpression of MEG3 increased overall survival, and overexpression of seven other lncRNAs (FGF14-AS2, AFAP1-AS1, EPB41L4A-AS2, BC040587, EGOT, GAS6-AS1, and FENDRR) had a positive effect on it [26]. This meta-analysis also showed a relationship between altered lncRNA expression and certain clinical-pathological factors. Overexpression of MALAT1 was significantly associated with the presence of a progesterone receptor, and overexpression of TUSC7 with the presence of a HER2 receptor. Overexpression of MEG3 correlated with a lower degree of histological malignancy G, overexpression of NEAT1 and TP73-AS1 with a higher degree of breast cancer according to the TNM classification [26]. In the above-described work, Tian et al. meta-analyzed 70 publications. The amount of data on the level of expression of lncRNA and their importance in breast cancer and other malignant tumors is constantly increasing. In the case of transcribed ultra-conservative regions, only two studies described in the introduction can be found. In the first, by Marini et al., uc.63 was determined in cultured breast cancer cell lines, and then, using a bioinformatics analysis, the prognostic significance of the transcript under study was assessed [27]. The subject of another study was the analysis of the expression of uc.38 also in cultured breast cancer cell lines, as well as in 100 of its samples [28]. In addition, the relationship between the expression of uc.38 and selected clinical-pathological factors was established, the effect of this expression on the breast cancer cell cycle was evaluated in vitro, a correlation between the expression of uc.38 and the function of the transcription factor PBX1 was found, and, finally, the effect of increased expression of uc.38 on the development of breast cancer in vivo was investigated [28].

The aim of the study was to determine the expression of long non-coding RNA uc.38 and uc.63 in female breast cancer patients in relation to control material and clinical-pathological features.

## 2. Materials and Methods

### 2.1. Patients

The study group consisted of 100 patients with infiltrating ductal breast cancer, operated on in the Department of Surgical Oncology and Breast Diseases Polish Mother’s Memorial Hospital-Research Institute (PMMH-RI) in Lodz. The exclusion criteria were: preoperative chemotherapy, preoperative hormone therapy, preoperative anti-HER2 therapy, or radiotherapy. The material for the research (test group) consisted of 100 paraffin blocks, containing excerpts from the above-described malignant cancerous tumors, collected in the archive of the Department of Pathology of the PMMH-RI in Lodz. The control group included 100 patients with benign breast tumors, operated on in the Department of Surgical Oncology and Breast Diseases of the PMMH-RI in Lodz. The material for the study consisted of 100 paraffin blocks, containing excerpts from the above-described benign breast changes, collected in the archive of the Department of Pathology of the PMMH-RI in Lodz. From the paraffin blocks, 2 scraps of 5 μm thick to 2 mL of Eppendorf tubes were taken. Histopathological and genetic tests were carried out in the Department of Pathology of the PMMH-RI and the Laboratory of Cancer Genetics of the PMMH-RI. The consent of the Bioethics Committee at the PMMH-RI in Lodz was obtained for the study on 26 February 2019—consent number 21/2019. Demographic characteristics of patients, clinical-pathological characteristics of preparations and information on the expression of ER, PR, and HER2 receptors are included in Table 1, Table 2 and Table 3, respectively.

### 2.2. Analysis of Expression of lncRNA uc.38 and uc.63 in Breast Cancer in Women

#### Isolation of RNA from Tissues Fixed in Paraffin

Isolation of total RNA from breast cancer preparations fixed in paraffin was carried out according to the method of Körbler et al. [29] in own modification. Paraffin from the tested samples (20–50 mg) was removed by successive rinses using xylene (4–8-fold) and ethyl alcohol (99.8%, 3–4-fold). The fragmented tissues were then suspended in 500 μL of buffer containing 10 mM NaCl, 500 mM Tris (pH 7.6), 20 mM EDTA, 1% SDS, and 500 μL/mL proteinase K and then incubated at 50 °C overnight. From the lysates obtained in this way, RNA was isolated using TRI Reagent (Sigma Aldrich, Darmstadt, Germany). After digestion of the tested preparations with proteinase K, 500 μL of TRI Reagent was added. After 10 min of incubation at room temperature, 200 μL of chloroform was added, and the whole was stirred for 30 s using a shaker. The samples were then centrifuged at 4 °C for 15 min at 12,000× *g*. The aqueous phase was transferred to a new tube, 500 μL of isopropanol was added to it and the whole was mixed. After 15 min of incubation at room temperature, the samples are centrifuged at 4 °C for 10 min at 12,000× *g*. The supernatant was removed and 1 mL of 75% ethanol was added to the RNA precipitate. The samples were then centrifuged again at 4 °C for 5 min at 12,000× *g*. After removing the supernatant, the sediment was dried at room temperature for 5 min. Then, 20 μL of water devoid of ribonuclease activity was added to the sediment using DEPC and the whole was mixed until the RNA sediment was completely dissolved. RNA samples were stored at −80 °C.

### 2.3. Spectrophotometric Analysis of Purity and RNA Concentration

The purity of the obtained RNA preparations was determined by spectrophotometric method by twice measuring the absorbance of each sample at wavelengths of 260 nm and 280 nm. The adopted criterion for RNA purity was A260/A280 within 1.8–2.0. The RNA concentration was determined by spectrophotometric method based on the absorbance value measured at a wavelength of 260 nm. This value corresponds to the following dependency:1OD = 40 µg RNA/mL(1)

### 2.4. Reverse Transcription Reaction

The reverse transcription reaction was performed using the High Capacity cDNA Reverse Transcription Kit (Applied Biosystems, St. Louis, MO, USA) in accordance with the manufacturer’s recommendations. 2 μg of RNA was suspended in 10 μL of DEPC treated water and then added 2 μL 10× RT Buffer, 0.8 μL dNTP (100 mM), 2 μL 10× RT Random Primers, 1 μL MultiScribe Reverse Transcriptase, and 4.2 μL of DEPC treated water. The 20 μL samples prepared in this way were incubated for 10 min at 25 °C, followed by 120 min at 37 °C and 5 min at 85 °C. The resulting cDNA was stored at −80 °C.

### 2.5. PCR Response with Real-Time Product Increment Analysis (Real Time PCR)

The Real Time PCR reaction was carried out using specific primers, the sequence of which was determined on the basis of data contained in the database UCbase 2.0, http://ucbase.unimore.it. For uc.38 these were sequences 5′–CCTTGAACCTGCTGGAAGAG-3′ and 5′-AACAGAGGGATGCTTTATTGC–3′ while for uc.63 the starter sequences were as follows 5′-CAGTGTTTGCCTGTTTGCTTGC-3′ and 5′-CCTGTTGCTTTCTTTCTGTTCCTC-3′. GAPDH was used as the reference gene, for which the primers had sequences 5′-GAGTCAACGGATTTGGTGGT-3′ and 5′-GACAAGCTTCCCGTTCTCAG-3′. The above primers amplified the following sequences (according to the data UCbase 2.0) presented in the Table 4.

In the Real Time PCR technique, the initial amount of the matrix is determined on the basis of the Ct (copy threshold) parameter—this is the theoretical number of the cycle in which the fluorescence value is higher than the arbitrarily assumed limit value. The reaction mixture contained 4 μL of SYBR^®^ Green JumpStart™ Taq Ready Mix™ (Sigma Aldrich, Darmstadt, Germany), 1 μL of cDNA, 0.5 μL of each pair of 5 μM primers, and 3 μL of RNAz-free water. The Real Time PCR reaction was carried out in the Mastercycler^®^ ep realplex (Eppendorf, Germany). The thermal profile of the reaction included pre-denaturation at 94 °C for 2 min, followed by 40 cycles involving incubation for 15 s at 94 °C and 1 min at 61 °C. The obtained Ct values were converted into the number of mRNA copies of the tested genes per 1000 copies of mRNA of the GAPDH gene according to the following relationship:ΔCt = Ct test gene − Ct reference gene;

L = 1000*2 − ΔCt.
Each procedure was repeated three times.

### 2.6. Statistical Analysis of Results

The normality of the distribution was analyzed by the Shapiro–Wolf test. The obtained results were statistically analyzed using the chi-squared independence test to compare frequencies or frequency distributions; the Mann–Whitney U test for the comparison of two independent groups, while the analysis of multiple comparisons of average ranks for all samples was performed using the Kruskal–Wallis test with the Dunn post hoc test χ. All test results were developed using PQStat v. 1.6.6 (PQStat Software, Poznan, Poland). All statistical tests were carried out at a significance level of α = 0.05.

## 3. Results

The mean expression values of the studied molecules and the results of the statistical analysis are presented in Table 5. As a result of the analyses, statistically lower expression of uc.38 was found in female ductal carcinoma preparations compared to expression in control material (*p* < 0.0001). In contrast, in the case of uc.63 expression in analogous material, the expression was statistically significantly higher in tumor material (*p* < 0.0001). The results of the analysis of uc.38 and uc.63 expressions depending on menopausal status are included in Table 6. As a result of the conducted analyses, it was shown that the expression of uc.63 is statistically significantly negatively correlated with the age of patients (Figure 1). Spearman’s r is −0.20 at *p* = 0.04. However, in the case of analyses of the expression of uc.38 and uc.63 in relation to the menopausal status of patients, no statistically significant relationships were found. Evaluation of uc.38 expression according to the clinical-morphological characteristics of ductal carcinoma of the breast showed the following statistically significant differences:With regard to the T trait, there was a statistically significant decrease in the expression of uc.38 in preparations classified as T3 in relation to the group of preparations classified as T2 (*p* < 0.05);With respect to the TNM stage, there was a statistically significant decrease in the expression of uc.38 in stage IV for stages I, II and III, *p* values < 0.05 in all three cases;With regard to the expression of ER, PR and HER2 receptors, statistically significantly lower expression of uc.38 was demonstrated in the group of preparations expressing all three receptors, compared to the group of preparations not expressing ER, PR, and HER2 receptors (*p* = 0.018).

Mean expression values of uc.38 depending on the analyzed morphological parameters and the results of statistical analysis are presented in Table 7. Evaluation of uc.63 expression according to the clinical-morphological characteristics of ductal carcinoma of the breast showed the following statistically significant differences:With regard to the M trait, a statistically significant decrease in the expression of uc.63 in preparations classified as M0 was observed in relation to the group of preparations classified as M1 (*p* = 0.036);With regard to the expression of the HER2 receptor, there was a statistically significant increase in the expression of uc.63 in preparations characterized by the expression of this receptor in relation to preparations not expressing this receptor, *p* value = 0.035.

Mean expression values of uc.63 depending on the analyzed morphological parameters and the results of statistical analysis are presented in Table 8.

## 4. Discussion

In the paper, an analysis of the expression of both uc.63 and uc.38 in the preparations of 100 patients with ductal breast cancer was carried out, compared to the control group, consisting of preparations of 100 patients with mild breast lesions. In addition, the relationship between the expression of the transcripts studied and clinical-pathological factors, such as age, menopausal status, individual elements of the TNM classification, the degree of advancement according to this classification, and the level of expression of steroid receptors and HER-2 receptor were evaluated. In order to compare the expressions of uc.63 and uc.38 in the study group and the control group, a statistical analysis was performed using the Mann–Whitney test. The analysis showed that uc.63 expression was statistically significantly higher in cancerous tissues compared to healthy tissues (*p* < 0.0001).

In the study by Marini et al. [27], expression of the transcript studied was determined in 12 breast cancer cell lines (MCF-7, T-47D, MDA MB 231, MDA MB 468, BT-20, BT-549, MDA MB 453, ZR-75-1, BT-474, SUM 149 PT, HCC1937, HCC1954) and compared to low expression of uc.63 in the Human Mammary Epithelial Cells cell line (HMEC), resembling normal breast tissue. In the obtained results, however, a significant discrepancy in the expression of uc.63 was found between individual cell lines. In six of them (MCF-7, T-47D, MDA MB 231, MDA MB 468, BT-20, BT-549) the expression of uc.63 was low. Four of these lines came from advanced breast cancer. Only two cells (MCF-7, T-47D) were found to express estrogen receptors, and in T-47D also progesterone receptors. In the six remaining breast cancer cell lines, uc.63 expression was high (MDA MB 453, ZR-75-1, BT-474, SUM 149 PT, HCC1937, HCC1954) [27]. Four of these lines came from early breast cancer. In the cells of one of them (ZR-75-1) there was expression of the estrogen receptor, in the next single line (HCC1954) and overexpression of the HER2 receptor. In order to carry out further stages of the experiment, including the effect of uc.63 on the XPO1 gene and the cell cycle, the authors selected the MDA MB 453 line with high uc.63 expression, derived from disseminated, triple-negative breast cancer. The authors did not investigate the relationship between uc.63 expression and clinical-pathological factors, but assessed the prognostic significance of high concentrations of the transcript in question. As a result of bioinformatics analysis of more than 2000 breast cancer preparations from The Cancer Genome Atlas, it was shown that increased expression of uc.63 in breast cancer tissues was associated with a significantly shorter DFS time in patients with luminal breast cancer A [27]. However, subgroup analysis after the PAM50 molecular assay did not reveal a significantly higher expression of uc.63, both in luminal carcinoma A tissues and in the tissues of other subgroups [27]. The authors interpreted this result as a correlation between high expression of uc.63 and greater aggressiveness of luminal breast cancer A and suggested that impaired expression of uc.63 could constitute a potential negative prognostic factor in this molecular subtype of the malignant tumor in question [27].

Statistical analysis of the relationship between the expression of uc.63 and selected clinical-pathological factors in the study was performed using the Kruskal–Wallis test with the Dunn post hoc test. As a result of the analysis, it was shown that the expression of uc.63 is statistically significantly negatively correlated with the age of patients (*p* = 0.04). A statistical relationship has not been shown for the menopausal state of the patients. Correlation analysis between the expression of uc.63 and other clinical-pathological features revealed a statistically significant decrease in the expression of the transcript in preparations with the M0 trait compared to preparations with the M1 trait (*p* = 0.036), which would suggest an association between the concentration of uc.63 and the tendency to relapse or aggressiveness of breast cancer. In the study, the relationship between high expression of uc.63 and overexpression of the HER2 receptor compared to preparations not showing the presence of the HER2 receptor (*p* = 0.035) was also statistically significant.

It is worth noting that in the only breast cancer cell line with overexpression of the HER2 receptor from the cited study by Marini et al., the expression of uc.63 was also high [27].

Statistical analysis did not show a significant correlation between the expression of uc.63 and the presence of steroid receptors. Perhaps molecular testing, such as PAM50 in the Marini et al. study, would allow a subgroup to be identified among luminal breast cancers in which high expression uc.63 could have prognostic significance.

To evaluate the expression of uc.38, the second transcript of the ultra-conservative region studied, the Mann–Whitney test was re-performed. On its basis, a statistically significant lower expression of uc.38 was found in the preparations of the study group compared to the preparations of the control group (*p* < 0.0001).

Expression of uc.38 was also determined in the study by Zhang et al. [28]. Uc.38 expression was evaluated not only in breast cancer samples compared to healthy breast tissue, but also in six cultured cell lines (MCF-7, ZR-75-1, BT474, MDA-MB-231, SUM1315, SK-BR-3) compared to breast epithelial cell lines (MCF-10A). Five of these lines came from advanced breast cancer (MCF-7, ZR-75-1, MDA-MB-231, SUM1315, and SK-BR-3), estrogen receptor expression (MCF-7) was present in one line, HER2 receptor expression (SK-BR-3) was present in the next. As a result of the analyzes, it was found that the expression of uc.38 was reduced both in the above cultured breast cancer cell lines (compared to the epithelial cell lines of the mammary gland) and in breast cancer preparations (compared to healthy breast tissue preparations) [28]. Correlation analysis between uc.38 expression and selected clinical-pathological factors subsequently indicated that decreased expression of uc.38 was associated with a larger diameter of the primary tumor, as well as a higher stage of cancer according to the TNM classification [28].

A similar trend can be observed in the results of presented study. Statistical evaluation using the Kruskal–Wallis test showed significantly lower expression of uc.38 in the preparations of the study group classified as T3 compared to the preparations designated as T2 (*p* < 0.05). In addition, there was a statistically significant decrease in the expression of uc.38 in preparations derived from disseminated breast cancer (stage IV according to the TNM classification) compared to the stages classified as I, II, and III (*p* < 0.05 in each case).

The authors of the Chinese study, based on the results obtained, suggested that uc.38 may be important in the control of tumor transformation and progression of breast cancer [28]. However, they did not mention any relationship between the expression of uc.38 and the expression of steroid receptors or the HER2 receptor.

In presented work, however, statistical analysis indicated a significantly lower expression of the transcript in preparations with the presence of all three above-mentioned receptors compared to preparations with triple-negative breast cancers (*p* = 0.018). In the presented study, an analysis of the relationship between the expression of uc.63 and the expression of uc.38 was also performed in both the study group and the control group. This correlation turned out to be positive and statistically significant in both of the above groups (control group-r = 0.25; *p* = 0.016; test group-r = 0.30; *p* = 0.003). The analysis of the correlation between the expression of the described transcripts was also investigated in relation to selected clinical-pathological features. A positive, statistically significant relationship between uc.63 and uc.38 was observed in the T1 breast cancer group (r = 0.46; *p* = 0.025), N0 (r = 0.27; *p* = 0.04), M0 (r = 0.30; *p* = 0.0034), as well as in the group with the lowest grade I (r = 0.45; *p* = 0.041) according to the TNM classification. The above relationship is not observed in the group of patients with a higher stage of breast cancer according to this classification.

To date, no paper has published a paper assessing the correlation between uc.63 expression and uc.38 expression in breast cancer.

The short time of observation and the experimental nature of the aforementioned works by Marini et al. [27] and Zhang et al. [28], whether the analysis carried out by me does not allow to clearly determine the significance of uc.63 and uc.38 in the initiation and course of breast cancer. However, they suggest their significant prognostic potential. In addition, it seems logical to assume, also based on the studies cited earlier [30,31] regarding the expression of selected lncRNAs in malignant tumor tissues, that uc.63 and uc.38 may play the role of oncogene and carcinogenesis suppressor, respectively.

Differentiated expression of uc.63 in cultured breast cancer cell lines described in the article by Marini et al. [27], the relationship between uc.38 and the transcription factor PBX1 [28] proved in the Chinese study, the effect of both molecules on the cell cycle observed in in vitro experiments [27,28], or the correlation between the molecules in question in less advanced cancers according to the TNM classification, which I have done, indicate, however, much more complicated functions of transcribed ultra-conservative regions. Specify these functions, or dependencies between uc. and neighboring or distant genes and their products certainly require further insightful experience.

## 5. Conclusions

The demonstrated relationship between the expression of uc.63 and uc.38 and the occurrence of breast cancer indicates their significant prognostic potential. The current state of knowledge on lncRNA in breast cancer is still limited. Further studies are warranted to further explore this subject.

## Figures and Tables

**Figure 1 genes-13-00608-f001:**
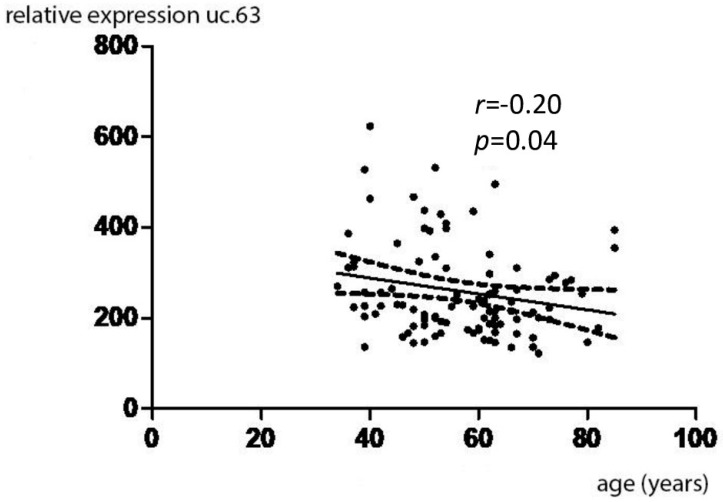
Expression of uc.63 in patients with breast carcinoma in relation to the age.

**Table 1 genes-13-00608-t001:** Demographic characteristics of breast cancer patients (test group) and control group.

	Test Group *n* (%)	Control Group *n* (%)	*p* ^a^
Group size Age, number of years <50 years ≥50 years Menopausal status pre peri post no data available	*n* = 100 56.76 ± 12.01 27 (27.0) 73 (73.0) 24 (24.0) 3 (3.0) 66 (66.0) 7 (7.0)	*n* = 100 59 ± 13.17 36 (36.0) 64 (64.0)	0.17

^a^ Test chi^2^.

**Table 2 genes-13-00608-t002:** Characteristics of the clinical-pathological features of patients.

	Variable	Test Group *n* (%)
TNM grade	T1	23(23.0)
	T2	57 (57.0)
	T3	17 (17.0)
	T4	3 (3.0)
	NX	1 (1.0)
	N0	55 (54.0)
	N1	25 (25.0)
	N2	13 (13.0)
	N3	6 (6.0)
	MX	3 (3.0)
	M0	92 (92.0)
	M1	5 (5.0)
Stage	I	19
	II	58
	III	18
	IV	5

**Table 3 genes-13-00608-t003:** Expression of ER, PR, and HER2 receptors in patients.

	Variable	Test Group *n* (%)
gene expression profile/IHC	luminal A ER+ PR+, HER2–	56 (56.0)
	luminal B ER+ PR+, HER2+	23 (23.0)
	basal-like ER–, PR–, HER2–	9 (9.0)
	HER2-like ER–, PR–, HER2+	12 (12.0)

**Table 4 genes-13-00608-t004:** Sequences of the amplified regions.

	Location	Sequence
uc.38	Chr 1:163939955-162206802 upstream CDCA1 downstream PBX1	CCTTGAACCTGCTGGAAGAGTTAGTATGTAAATTTCAACCTATTTTTAAGGGTTATTTTCACTCAAGTGAAATCTATCAAGGAAGGAGGGTTATTTTTACAGCATACAGCAACTGCTGATCACCATGGCAACCGGCCTGGTGAAATGCAATAAAGCATCCCTCTGTTATCGTAAACACAAAAGGGAAACACTGAAATCTAAAAGAAAGGCAATTATTGTAG
uc.63	Chr 2:61752501-617552778 upstream USP34 downstream AK023367	TGATTAATATCAGGTAGTGTTAACATCTTTAAAGAAAAAAAAATGATTGCATAAAAGCCAAATGTCATAGTGCATAAATTTAGCACCAAATCATTTGTAATTTATGTAAATTGAAGAATTCTTTACCTGTTGCTTTCTTTCTGTTCCTCTAATCATCTCATTTTTCACAAGACAAATTTGAGTTTTTAAAAATACTGTTGATAAATCAACTTAAACATTAGTAATGTCTGTCAGTATAAAAAGCAAAATTTACCAGGCAAGCAAACAGGCAAACACTG
*GAPDH*		GAGTCAACGGATTTGGTCGTATTGGGCGCCTGGTCACCAGGGCTGCTTTTAACTCTGGTAAAGTGGATATTGTTGCCATCAATGACCCCTTCATTGACCTCAACTACATGGTTTACATGTTCCAATATGATTCCACCCATGGCAAATTCCATGGCACCGTCAAGGCTGAGAACGGGAAGCTTGTC

**Table 5 genes-13-00608-t005:** Expression of uc.38 and uc.63 in breast ductal carcinoma patients and control.

uc.	Test Material	Median	Percentile 25	Percentile 75	*p* ^a^
uc.38	Carcinoma	700.91	574.49	921.24	<0.0001
	Control	1262.788	866.84	1786.67	
uc.63	Carcinoma	226.85	186.47	310.51	<0.0001
	Control	133.62	97.18	168.92	

^a^ Mann–Whitney U test.

**Table 6 genes-13-00608-t006:** Expression of uc.38 and uc.63 in breast cancer patients in relation to menopausal status.

uc.	Menopausal Status	Median	Percentile 25	Percentile 75	*p* ^a^
u.38	Pre	664.31	472.99	788.95	>0.05
	Peri	1012.44	966.09	1036.32	
	Post	700.90	575.71	908.68	
uc.63	Pre	255.03	188.68	323.74	>0.05
	Peri	228.68	197.74	318.97	
	Post	217.51	185.19	274.58	

^a^ Kruskal–Wallis test.

**Table 7 genes-13-00608-t007:** Expression of uc.38 in breast cancer patients depending on clinical-pathological factors.

Feature		Median	Percentile 25	Percentile 75	*p*
T	T1	666.77	544.95	881.92	>0.05
	T2	694.29	605.97	972.90	T1 vs. T2 >0.05
	T3	481.38	324.74	757.17	T1 vs. T3 >0.05
	T4	719.82	718.39	721.24	T1 vs. T4 >0.05
					T2 vs. T3 <0.05
					T2 vs. T4 >0.05
					T3 vs. T4 >0.05
N	N0	685.58	573.74	884.88	>0.05
	N1	690.81	470.99	853.45	
	N2	974.28	698.12	1134.43	
	N3	700.93	558.62	747.12	
M	M0	696.21	574.49	930.85	>0.05
	M1	716.97	572.65	816.56	
stage	I	666.78	574.11	903.44	0.0085
	II	690.81	573.03	877.57	I vs. II >0.05
	III	764.54	658.77	1038.85	I vs. III >0.05
	IV	216.97	216.56	478.08	I vs. IV <0.05
					II vs. III >0.05
					II vs. IV <0.05
					III vs. IV <0.01
ER	ER+	712.73	573.74	941.13	>0.05
	ER−	690.81	599.64	831.36	
PR	PR+	716.97	574.87	925.72	>0.05
	PR−	686.80	583.94	876.78	
HER2	HER2+	642.83	511.31	806.42	>0.05
	HER2−	684.62	574.49	848.23	
all	all+	424.31	351.80	496.82	0.018
	all−	684.89	611.90	786.72	

**Table 8 genes-13-00608-t008:** Expression of uc.63 in breast cancer patients depending on clinical-pathological factors.

Feature		Median	Percentile 25	Percentile 75	*p* ^a^
T	T1	238.76	178.09	312.67	>0.05
	T2	224.47	187.10	297.39	
	T3	239.62	215.31	323.33	
	T4	217.86	207.29	228.43	
N	N0	224.82	168.75	282.96	>0.05
	N1	252.35	200.50	365.00	
	N2	215.31	178.45	242.48	
	N3	220.67	201.51	250.95	
M	M0	227.91	187.51	310.51	0.036
	M1	778.45	769.12	846.26	
Stage	I	253.32	180.68	312.67	>0.05
	II	690.81	573.03	877.57	
	III	233.07	200.13	257.67	
	IV	178.45	169.11	196.73	
ER	ER+	227.10	185.33	307.42	>0.05
	ER−	226.65	192.29	294.47	
PR	PR+	227.15	187.65	323.33	>0.05
	PR−	225.48	185.81	290.92	
HER2	HER2+	281.48	265.70	311.86	0.035
	HER2−	220.87	186.47	258.51	
all	all+	261.25	256.80	265.70	>0.05
	all−	219.16	187.10	253.80	

^a^ Kruskal–Wallis test.

## Data Availability

All data and materials, as well as software application, support the published claims and comply with field standards.

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
