# Peer review of "Analysis of Long Non-Coding RNA (lncRNA) uc.38 and uc.63 Expression in Breast Carcinoma Patients"

_genes, 2022, doi:10.3390/genes13040608_

Round 1

Reviewer 1 Report

This interesting study revealed the relationship between the expression of uc.63 and uc.38 and the 380 occurrence of breast cancer indicates their significant prognostic potential. Meanwhile, statistical dependency analysis of the expression of uc.63 and uc.38 and the selected clinical and pathological factors showed, that the expression of uc.63 statistically drops with the patient’s age, is higher in the breast cancer tissue type M1 according to the TNM classification and in tissues with overexpressed HER2. Albeit, I consider these findings to provide insight into the development of cancer research, I still have some minor suggestions.

1, It would be much better if the authors could provide either Workflow or Graphical abstract for this manuscript, so that readers can easily understand the concept of the current analysis.

2, For the Statistical analysis of results, the author used the Mann-Whitney U test for the comparison of two independent groups, while the analysis of multiple comparisons of average ranks for all samples was performed using the Kruskal-Wallis test with the Dunn posthoc test. Please explain and described more detail of the rationale and method in this section.

3, From table 1 to table 7, did the author perform univariate or multivariate comparisons?

4, There are several errors in grammar and punctuation: for example, "P value". ("P" capital letter in italic style). Table 3, ER+PR+HER2+, + should be upper index.

5, Please edit table legends again, and the manuscript needs proofreading.

Author Response

Thank you for your review.

I would like to kindly ask you to reconsider the publication of our revised paper:

Analysis of long non-coding RNA (lncRNA) uc.38 and uc.63 expression in breast carcinoma patients

I hereby provide responses to the reviewers and list the changes that have been made in the revised version of our paper.

 1, It would be much better if the authors could provide either Workflow or Graphical abstract for this manuscript, so that readers can easily understand the concept of the current analysis.

Graphical abstract has been created .

2, For the Statistical analysis of results, the author used the Mann-Whitney U test for the comparison of two independent groups, while the analysis of multiple comparisons of average ranks for all samples was performed using the Kruskal-Wallis test with the Dunn posthoc test. Please explain and described more detail of the rationale and method in this section.

3, From table 1 to table 7, did the author perform univariate or multivariate comparisons?

Statistical analyses were carried out by an expert statistician. The normality of the distribution was analyzed by the Shapiro-Wolf test.  The obtained results were statistically analyzed using  chi-squared test (2) independence for comparison of frequencies or frequency distributions; c the Mann-Whitney U test to compare two independent groups, while the analysis of multiple average-rank comparisons for all trials was performed using the Kruskal-Wallis test with the Dunn post-hoc test. All test results were developed using PQStat v. 1.6.6 (PQStat Software, Poland).  All statistical tests were carried out at a significance level of α = 0.05.

4, There are several errors in grammar and punctuation: for example, "P value". ("P" capital letter in italic style). Table 3, ER+PR+HER2+, + should be upper index.

Table 3 has been revised in accordance with the recommendations of the second appointed reviewer. Errors have been corrected

5, Please edit table legends again, and the manuscript needs proofreading.

Has been corrected

I hope you find our revised Manuscript satisfying so that it can meet the criteria of publication in your Journal.

Looking forward to hearing from you,

Yours sincerely,

Beata Smolarz

Reviewer 2 Report

In the manuscript entitled "Analysis of long non-coding RNA (lncRNA) uc.38 and uc.63 expression in breast carcinoma patients" the authors determined the expression of long non-coding RNA uc.38

and uc.63 in patients with female breast cancer in relation to the control material and clinical pathological characteristics.

Various studies have shown how lncRNAs are involved in tumorigenesis. however it has not yet been determined how T-UCRs contribute to the initiation of carcinogenesis, so the study is interesting but there are some suggestions.

1). In the "Introduction" section the authors cite the classification of Jarroux J et al. for lncRNAs (title to be corrected in references - History, Discovery, and Classification of lncRNAs. Jarroux J, Morillon A, Pinskaya M. Adv Exp Med Biol. 2017; 1008: 1-46). They should explain T-UCR sequences. Not all readers may have the appropriate skills and knowledge. We suggest the reference "Mestdagh P et al., An integrative genomics screen discovers the functions of ncRNA T-UCR in neuroblastoma tumors. Oncogene. 17 Jun 2010; 29 (24): 3583-92.".

2). In the section “Patients - Materials and Methods”, the authors report the consensus number for the ethics committee dated 2019. Are the 200 samples analyzed for the study referred to patients selected in the same year?

3). Table 3 shows the receptor expression (ER, PR and HER2), as individual markers. We suggest to report the “gene expression profile/IHC” classification: Luminal A, Luminal B, Basal Like, Her2-Like.

4). From Tables 4 and 5, referring to the results obtained respectively with the Mann-Whitney U test and with the Kruskal-Wallis test. It’s not clear or it hasnt been reported how the expression of the lncRNA-uc under examination is altered according to age of patients. The authors in the "Discussion" section indicate (Line 303-304) that "the expression of uc.63 is statistically significantly negatively correlated with the age of the patients (p = 0.04)". The table in this manuscript may mislead readers. We suggest reporting the data.

Author Response

Thank you for your review.

I would like to kindly ask you to reconsider the publication of our revised paper:

Analysis of long non-coding RNA (lncRNA) uc.38 and uc.63 expression in breast carcinoma patients

I hereby provide responses to the reviewers and list the changes that have been made in the revised version of our paper.

1). In the "Introduction" section the authors cite the classification of Jarroux J et al. for lncRNAs (title to be corrected in references - History, Discovery, and Classification of lncRNAs. Jarroux J, Morillon A, Pinskaya M. Adv Exp Med Biol. 2017; 1008: 1-46). They should explain T-UCR sequences. Not all readers may have the appropriate skills and knowledge. We suggest the reference "Mestdagh P et al., An integrative genomics screen discovers the functions of ncRNA T-UCR in neuroblastoma tumors. Oncogene. 17 Jun 2010; 29 (24): 3583-92.".

It has been corrected

2). In the section “Patients - Materials and Methods”, the authors report the consensus number for the ethics committee dated 2019. Are the 200 samples analyzed for the study referred to patients selected in the same year?

Yes, in the same year

). Table 3 shows the receptor expression (ER, PR and HER2), as individual markers. We suggest to report the “gene expression profile/IHC” classification: Luminal A, Luminal B, Basal Like, Her2-Like.

Table 3 has been corrected in line with the recommendations

4). From Tables 4 and 5, referring to the results obtained respectively with the Mann-Whitney U test and with the Kruskal-Wallis test. It’s not clear or it hasnt been reported how the expression of the lncRNA-uc under examination is altered according to age of patients. The authors in the "Discussion" section indicate (Line 303-304) that "the expression of uc.63 is statistically significantly negatively correlated with the age of the patients (p = 0.04)". The table in this manuscript may mislead readers. We suggest reporting the data.

We agree with the review. The description has been corrected. The results have been presented on Figure 1 .

I hope you find our revised Manuscript satisfying so that it can meet the criteria of publication in your Journal.

Looking forward to hearing from you,

Yours sincerely,

Beata Smolarz